# Angiogenesis in Atrial Fibrillation: A Literature Review

**DOI:** 10.3390/biomedicines13061399

**Published:** 2025-06-06

**Authors:** Jie Lin, Haihuan Lin, Zhijun Xu, Zhihui Yang, Chenglv Hong, Ying Wang, Haocheng Lu

**Affiliations:** 1Department of Cardiology, The First Affiliated Hospital of Wenzhou Medical University, Wenzhou 325000, China; wzmclinjie@126.com (J.L.); wzmcxiaohuan@126.com (H.L.); zhihui_yang99@163.com (Z.Y.); hongchenglv@wmu.edu.cn (C.H.); 2School of Pharmaceutical Sciences, Guangdong University of Chinese Medicine, Guangzhou 510006, China; m18302084439@163.com; 3Department of Pharmacology, Joint Laboratory of Guangdong-Hong Kong Universities for Vascular Homeostasis and Diseases, SUSTech Homeostatic Medicine Institute, School of Medicine, Southern University of Science and Technology, Shenzhen 518055, China

**Keywords:** angiogenesis, atrial fibrillation, VEGF, fibrosis, therapeutic targeting

## Abstract

Atrial fibrillation (AF), the most prevalent clinically significant cardiac arrhythmia, is characterized by chaotic atrial electrical activity and currently affects an estimated 2.5–3.5% of the global population. Its pathogenesis involves ion channel dysfunction, inflammatory cascades, and structural remodeling processes, notably fibrosis. Angiogenesis, the physiological/pathological process of new blood vessel formation, plays a multifaceted role in AF progression. This review synthesizes evidence highlighting angiogenesis’s dual role in AF pathogenesis: while excessive or dysregulated angiogenesis promotes atrial remodeling through fibrosis, and electrical dysfunction via VEGF, ANGPT, and FGF signaling pathways, compensatory angiogenesis exerts protective effects by improving tissue perfusion to alleviate ischemia and inflammation. Therapeutically, targeting angiogenic pathways—particularly VEGF—represents a promising strategy for modulating structural remodeling; however, non-selective VEGF inhibition raises safety concerns due to cardiovascular toxicity, necessitating cautious exploration. Emerging evidence highlights that anti-cancer agents (e.g., ibrutinib, bevacizumab) impair endothelial homeostasis and elevate AF risk, underscoring the need for cardio-oncology frameworks to optimize risk–benefit ratios. Preclinical studies on angiogenesis inhibitors and gene therapies provide mechanistic insights, but clinical validation remains limited. Future research should prioritize elucidating mechanistic complexities, developing biomarker refinement, and implementing interdisciplinary strategies integrating single-cell sequencing with cardio-oncology principles. This review emphasizes the imperative to clarify angiogenic mechanisms, optimize therapeutic strategies, and balance pro-arrhythmic versus compensatory angiogenesis, in pursuit of personalized AF management.

## 1. Introduction

Atrial fibrillation (AF), the most common clinically significant cardiac arrhythmia, is characterized by rapid, irregular atrial contractions resulting from chaotic electrical activity in the atria [1,2,3]. Affecting millions globally, its prevalence exhibits an age-dependent pattern, with current estimates of 2.5–3.5% of the global population [4,5]. While the incidence remains below 1% in individuals under 60 years of age, it escalates dramatically to 9% by age 80 [6]. This epidemiological trajectory parallels global population aging trends, with a doubling of AF-related disability-adjusted life years between 1990 and 2019, particularly in middle-income nations [7,8]. Notable demographic disparities exist, with a higher incidence observed in males and individuals of European descent [9]. Projections also highlight an impending public health challenge in China: 5.2 million men and 3.1 million women aged over 60 are anticipated to develop AF by 2050 [10]. Clinically, AF frequently coexists with cardiovascular comorbidities such as hypertension and coronary artery disease [4,11].

AF confers a 5-fold increase of stroke risk primarily mediated by left atrial thrombus formation, accounting for one-fifth of acute ischemic strokes [12]. This arrhythmia is independently associated with a 4–5-fold higher incidence of thromboembolic events, doubled mortality rates, and substantial contributions to heart failure development through tachycardia-mediated cardiomyopathy [6]. Symptomatic patients commonly experience clinically impactful quality of life impairments due to palpitations, exertional dyspnea, and fatigue, whereas asymptomatic cases often evade diagnosis until catastrophic complications manifest [7,13].

The pathogenesis of AF is orchestrated by a dynamic interplay of molecular perturbations, structural remodeling, and genetic predispositions. Electrical abnormalities, particularly ectopic foci predominantly arising from the pulmonary veins, are widely regarded as the primary instigators of AF [14]. Ion channel dysregulation—encompassing Na^+^, Ca^2+^, and K^+^ channels—drives dysregulated atrial electrical conduction, serving as a cornerstone of AF pathophysiology [15]. Notably, elevated serum levels of inflammatory markers correlate strongly with all-cause mortality and adverse cardiovascular outcomes in AF patients [16]. Comorbidities such as hypertension, coronary artery disease, obesity, and diabetes synergistically exacerbate AF progression via endothelial dysfunction, chronic inflammatory, and atrial dilatation. Histopathological alterations like myocardial fibrosis constitute the structural for AF initiation and perpetuation [17]. Furthermore, atrial electrical and autonomic remodeling critically sustain the arrhythmogenic milieu [4,18].

At present, the treatment for AF includes medications and cardioversion therapy. Additionally, surgical or catheter ablation may be required for patients refractory to medical treatment, though post-procedural AF recurrence remains a clinical challenge [19]. Existing therapeutic options targeting the underlying mechanical and structural changes in AF remain limited [20], highlighting the pressing need to deepen our understanding of AF pathogenesis and to develop innovative therapies to enhance patient outcomes.

Angiogenesis, the process of new blood vessel formation, exhibits dual roles in cardiovascular pathophysiology. While physiological/compensatory angiogenesis (which produces functional, organized, mature vessels with proper perfusion) facilitates tissue repair and improves perfusion [21,22,23], dysregulated angiogenesis (which produces abnormal leaky vessels with poor perfusion, usually due to imbalanced signaling) promotes fibrosis, inflammation, and electrical instability—key pathological mechanisms of AF progression [24,25]. For instance, vascular endothelial growth factor (VEGF), a master regulator of angiogenesis, demonstrates paradoxical roles in both compensatory vascular adaptation and pathological remodeling during AF [21,23,24]. This review seeks to systematically review the intricate relationship between angiogenesis and AF pathogenesis, with a focus on its context-dependent mechanisms and therapeutic implications.

## 2. Pathophysiological Mechanisms

Angiogenesis, defined as the formation of new vessels originating from existing vasculature, serves as a critical biological mechanism underlying embryonic morphogenesis, wound healing, and tissue homeostasis. This process is tightly regulated through a dynamic balance between pro-angiogenic mediators (e.g., vascular endothelial growth factor [VEGF], fibroblast growth factor [FGF]) and inhibitory counterparts like endostatin [11,26,27]. Under physiological conditions, angiogenesis facilitates tissue remodeling through two principal mechanisms: (1) supporting cellular proliferation during reproductive cycles via endometrial vascular expansion [28,29] and (2) restoring tissue perfusion post-ischemia through collateral vessel formation [30].

Pathological angiogenesis manifests across diverse disease spectra, including oncogenesis, chronic inflammatory disorders, and cardiovascular pathologies such as AF [4,26,31,32,33,34]. Within tumor microenvironments, neoplastic and stromal cells cooperatively secrete angiogenic regulators, with VEGF emerging as the predominant driver of tumor-associated vascular proliferation. This aberrant vascularization sustains malignant progression through dual mechanisms: ensuring nutrient and oxygen supply while establishing metastatic conduits.

Notably, AF has been mechanistically linked to angiogenesis dysregulation. Emerging evidence implicates cardiac hypoxia-induced VEGF overexpression and endothelial progenitor cell (EPC) dysfunction in AF pathogenesis. The resultant vascular permeability alterations and endothelial integrity compromise may perpetuate arrhythmogenic substrate formation through three pathways: (1) myocardial fibrosis via endothelial-to-mesenchymal transition, (2) microvascular rarefaction exacerbating electrical heterogeneity, and (3) inflammatory cascade activation [4,29,30,35,36]. Paradoxically, controlled angiogenesis demonstrates therapeutic potential through EPC-mediated endothelial repair mechanisms [37], suggesting context-specific biological roles in AF progression.

Collectively, these findings highlight angiogenesis as a double-edged sword in cardiovascular physiology—essential for tissue repair yet potentially detrimental when dysregulated. The intricate balance between pro- and anti-angiogenic factors warrants further investigation as a therapeutic target in AF management. In this review, we aim to systematically evaluate the multifaceted roles of angiogenesis in AF pathogenesis through a critical analysis of mechanistic studies and clinical evidence.

### 2.1. Angiogenesis and Atrial Electrical Remodeling

Angiogenesis serves as a compensatory mechanism to restore atrial myocardial perfusion by augmenting microvascular networks, thereby mitigating ischemic injury and oxygen deprivation in atrial fibrillation. Histopathological analyses reveal that elevated endothelial cell proliferation and microvascular density (MVD) in AF tissues suggest adaptive responses to chronic hypoxia and heightened metabolic demands caused by rapid atrial activation [29,38]. However, this reactive vascular remodeling demonstrates paradoxical inefficiency: while angiogenic markers (e.g., VEGF, HIF-1α) are upregulated, the resultant neovasculature often displays structural immaturity, such as impaired barrier function [39], leading to persistent tissue hypoxia and progressive fibrosis [30,40]. Despite localized increases in vessel diameter, overall MVD remains unchanged due to simultaneous microvascular dropout caused by endothelial-to-mesenchymal transition (EndMT) [30,41].

The interplay between angiogenesis and electrical remodeling is context dependent. Compensatory angiogenesis may transiently alleviate ischemia by improving perfusion, yet structurally immature vessels fail to resolve hypoxia [42], as evidenced by persistent HIF-1α stabilization and elevated carbohydrase IX levels in atrial biopsies [30]. Conversely, dysregulated angiogenesis exacerbates electrical heterogeneity through two key mechanisms: (1) Microvascular disorganization: Excessive angiogenesis generates leaky, non-functional microvessels that disrupt the atrial substrate, slowing conduction velocities and promoting reentrant circuits [38,39]; (2) Inflammatory amplification: Dysregulated angiogenesis recruits pro-inflammatory cells (e.g., macrophages) that release cytokines (e.g., IL-6, TNF-α), further driving fibrosis and oxidative stress [43,44].

VEGF-A overexpression enhances endothelial permeability and fibroblast activation, creating arrhythmogenic substrates through collagen deposition and interstitial fibrosis [4]. For instance, in preclinical models, VEGF-D overexpression initially improves perfusion but later exacerbates fibrosis via EndMT, illustrating the dual role of the angiogenic factor [31]. These pathological changes are compounded by reduced NO bioavailability and shear stress due to disorganized atrial contraction in AF, which further impairs endothelial function and perpetuates electrical instability [30,41].

### 2.2. Interplay Between Angiogenesis, Fibrosis, and Inflammation

Compensatory angiogenesis may reduce fibrosis and inflammation by improving vascular integrity and reducing oxidative stress [32,34,45]. For example, VEGF-B promotes endocardial vessel development and cardiac regeneration, suggesting a protective role in ischemic conditions [46]. However, abnormal angiogenesis driven by vascular endothelial growth factor (VEGF) can also lead to atrial remodeling, fibrosis, and inflammation, which are key pathological drivers in AF [47,48]. Elevated VEGF levels are associated with increased endothelial permeability, inflammation, and fibrosis, thereby potentially promoting AF progression [43,49]. Higher VEGF-A levels in AF patients may promote tissue remodeling and fibrosis, exacerbating the arrhythmogenic substrate [4].

While VEGF-A is widely recognized for its pro-fibrotic effects mediated by fibroblast activation and collagen deposition [50,51], emerging evidence suggests that VEGF-B enhances endothelial nitric oxide synthase (eNOS) activity through AMPK-mediated phosphorylation, thereby improving vascular function and attenuating oxidative stress [52]. Experimental studies demonstrate this mechanism improves cardiac protection against ischemic injury by increased NO production while reducing superoxide and malondialdehyde levels [53]. This functional dichotomy highlights the importance of isoform-specific targeting in AF management.

The interplay between inflammatory pathways and angiogenesis is mediated through several key cytokines and growth factors: (1) TGF-β pathway: TGF-β not only drives fibroblast-to-myofibroblast differentiation but also modulates angiogenesis by suppressing VEGF expression in hypoxic conditions [54]. This dual role creates a feed-forward loop where fibrosis impairs angiogenesis, further exacerbating tissue hypoxia. (2) FGF signaling: Fibroblast growth factor-2 (FGF-2) synergizes with vascular endothelial growth factor (VEGF) to promote endothelial cell proliferation and vascular network maturation [55]. (3) IL-6/STAT3 axis: Interleukin-6 (IL-6) upregulates VEGF expression via STAT3 phosphorylation, linking systemic inflammation to localized vascular remodeling [56].

### 2.3. Interaction Between Atrial Myocytes and Angiogenesis

The interaction between atrial myocytes and angiogenic processes is critical for maintaining cardiac function and preventing arrhythmias. Atrial myocytes can secrete various angiogenic factors, such as VEGF and fibroblast growth factor (FGF), which stimulate endothelial cell proliferation and neovascularization. Conversely, newly formed blood vessels modulate the metabolic environment of atrial myocytes, potentially improving their functional integrity and survival. However, in AF, this homeostatic crosstalk is often disrupted. For instance, increased fibrosis compromises myocyte–endothelial communication, leading to diminished angiogenesis and aggravation of the arrhythmic substrate [57].

Endothelial-to-mesenchymal transition underlies microvascular rarefaction in AF. During EndMT, endothelial cells lose their characteristic markers (e.g., VE-cadherin) and acquire mesenchymal phenotypes (e.g., α-SMA), resulting in vascular destabilization and compromised perfusion capacity [58]. This process is driven by pro-fibrotic cytokines such as TGF-β and mechanical stress from aberrant atrial contractility [59]. Key consequences of EndMT in AF include fibrotic remodeling. EndMT-transformed endothelial cells directly promote atrial interstitial fibrosis by secreting extracellular matrix (ECM) components (e.g., collagen I/III). Mechanistically, this process is central to TGF-β-induced EndMT pathogenesis, yet remains partially reversible through therapeutic interventions such as Sema3A activation or miR-181b inhibition [59,60,61].

### 2.4. Pathological Roles of Key Signaling Pathways

#### 2.4.1. VEGF Pathway: Promotes Fibrosis and Endothelial Dysfunction

VEGF promotes endothelial cell survival, proliferation, and migration while increasing endothelial permeability [26,34]. Elevated VEGF levels (especially VEGF-A and VEGF-D) are consistently observed in AF patients, demonstrating a strong association with AF risk [4,34,62]. These elevated levels are attributed to atrial enlargement and the increased mechanical stretch in AF [30,34]. VEGF promotes fibrosis by interacting with fibroblasts and myofibroblasts, and conversely, fibrosis exacerbates AF by disrupting myocardial bundle continuity and impairing local conduction [63]. In AF, VEGF and its receptor KDR (VEGFR2) are significantly upregulated, yet KDR nuclear translocation is reduced, suggesting compromised angiogenic signaling [30].

Beyond fibrosis, VEGF-A signaling disrupts endothelial barrier integrity through FAK-mediated phosphorylation of β-catenin at tyrosine-142, which induces dissociation of VE-cadherin/β-catenin complexes and junctional breakdown. This process increases vascular permeability and inflammatory cell infiltration, as evidenced by genetic and pharmacological FAK inhibition studies [64]. The resultant vascular leakage may exacerbate oxidative stress and create a pro-thrombotic microenvironment, though experimental validation is required to establish direct links to atrial electrical destabilization.

#### 2.4.2. ANGPT Pathway: Ang-2 Imbalance Leads to Endothelial Destabilization

Angiopoietins (Ang-1 and Ang-2) are critical for vascular stability and remodeling. Ang-1 promotes endothelial stabilization through Tie-2 receptor activation, whereas Ang-2 acts as its functional antagonist [65]. Elevated serum Ang-2 levels in AF patients correlate with endothelial dysfunction and thrombogenic propensity [34]. Dysregulation of this angiopoietin balance compromises vascular integrity, potentially contributing to AF pathogenesis [66].

#### 2.4.3. FGF/EGF Pathway: Regulates Fibrosis and Oxidative Stress

Atrial fibrosis, a hallmark of AF progression, is regulated by fibroblast growth factor-2 (FGF-2) through dual roles in angiogenesis and fibrogenesis. Mechanistically, FGF-2 upregulates VEGF in endothelial cells, driving angiogenesis while potentially influencing AF pathogenesis [55,67,68]. Clinical studies reveal serum FGF-23 levels are positively associated with AF risk [69,70], whereas FGF-21 demonstrates anti-fibrotic and antioxidant properties that attenuate AF progression [71].

EGF signaling modulates endothelial cell proliferation and differentiation, thereby influencing vascular homeostasis in AF [66]. Elevated circulating EGFR ligands—including EGF and heparin-binding EGF (HB-EGF)—are detected in AF patients [72], suggesting their potential role in AF-associated vascular remodeling.

#### 2.4.4. TGF-β/PDGF Pathway: Core Pro-Fibrotic Mechanism

TGF-β is a well-established pro-fibrotic cytokine that serves as a central regulator in cardiac remodeling and fibrosis. Chronically upregulated in AF patients, TGF-β mediates myofibroblast differentiation, thereby driving collagen production and extracellular matrix remodeling [73].

Functionally complementary to TGF-β, platelet-derived growth factor (PDGF) is another critical regulator governing cardiac fibrosis and remodeling. PDGF stimulates fibroblast proliferation and collagen synthesis, with its expression level directly correlating with the severity of atrial remodeling in AF patients [74].

## 3. Clinical Evidence of Angiogenesis Implication in AF

### 3.1. Association of Angiogenic Biomarkers with Atrial Fibrillation

The clinical assessment of angiogenesis biomarkers has gained significant attention in recent years due to their potential implications in cardiovascular diseases, including atrial fibrillation (AF). Biomarkers such as VEGF, galectin-3, and semaphoring 4D have been extensively studied for their roles in angiogenesis and AF pathophysiology. Elevated VEGF levels demonstrate a positive correlation with AF severity and atrial remodeling, supporting its utility as a biomarker for disease progression [75]. Similarly, galectin-3, a mediator of inflammation and fibrosis, is implicated in AF pathogenesis, highlighting dual roles as both a therapeutic target and fibrosis biomarker [73]. Semaphorin 4D emerges as a novel biomarker predictive of AF recurrence post-catheter ablation, underscoring its clinical relevance in outcome prognostication [76]. Table 1 summarizes key angiogenic pathways and their implications in AF pathogenesis.

### 3.2. Controversy over AF Risk Associated with Anti-Angiogenic Drugs

Anti-angiogenic drugs inhibit VEGF signaling to suppress pathological angiogenesis. Emerging evidence suggests these agents may paradoxically increase AF by disrupting angiogenic–inflammatory homeostasis. For instance, tyrosine kinase inhibitors (TKIs) such as sorafenib exhibit elevated AF incidence when combined with chemotherapy [78]. Proposed mechanisms include altered cardiac electrophysiology via endothelial dysfunction or oxidative stress induction [79]. Contrastingly, a nested case-control study in Taiwan involving cancer patients exposed to angiogenesis inhibitors (e.g., bevacizumab, sorafenib) demonstrated no significant AF risk association [28].

### 3.3. Dual Role of VEGF-D

While elevated VEGF-D levels are linked to incident AF and ischemic stroke in population studies [36], targeted therapeutic approaches (e.g., AdVEGF-DΔNΔC gene therapy) demonstrate enhanced myocardial perfusion without elevating arrhythmia risk [77,80]. This discrepancy likely stems from isoform-specific biological effects or variations in therapeutic administration strategies.

### 3.4. Mechanisms of Atrial Fibrillation Induced by Anticancer Drugs

Anthracyclines are well-documented for their cardiotoxic effects, which may induce structural and electrical cardiac remodeling, thereby elevating AF risk. For instance, doxorubicin induces oxidative stress and inflammation, driving atrial remodeling characterized by fibrosis and electrical instability that predispose patients to AF [81].

Fluorouracil (5-FU), another chemotherapeutic agent, is linked to AF development primarily through vasospasm and endothelial injury mechanisms, which may provoke ischemia and subsequent arrhythmogenesis [82]. Tyrosine kinase inhibitors, particularly ibrutinib, exhibit a markedly elevated AF incidence, with studies reporting up to 38% of patients developing this arrhythmia [83]. Although the precise mechanisms remain incompletely elucidated, proposed pathways including drug-induced autonomic dysregulation and enhanced atrial susceptibility to AF.

VEGF inhibitors (e.g., bevacizumab) compromise endothelial repair by suppressing EPC mobilization, thereby exacerbating microvascular rarefaction and atrial fibrosis [84,85]. Preclinical studies reveal that VEGF-induced vascular leak disrupts intercalated disc nanostructure—including connexin-43-mediated coupling—and sodium channel (NaV1.5) localization, resulting in slowed conduction velocity and facilitated reentry circuit formation [39].

### 3.5. Controversy over Cardiovascular Toxicity in Anti-Angiogenic Therapy

Anti-VEGF therapies, while primarily targeting tumor angiogenesis, carry cardiovascular complication risks, though their direct association with AF remains incompletely defined [86]. Contrastingly, angiogenesis inhibitors showed no significant new-onset AF risk correlation in the context of cancer treatment [28].

Sunitinib, a VEGFR2 inhibitor, demonstrated dose-dependent cardiotoxicity in a phase I/II trial involving advanced/metastatic renal cell carcinoma patients [87]. Comparative pharmacovigilance analyses identified higher cardiovascular/cerebrovascular adverse event rates with ranibizumab versus other anti-VEGF agents [88], emphasizing the need to consider systemic safety evaluations in therapeutic selection. Notably, patients with pre-existing endothelial dysfunction (e.g., diabetes or hypertension) exhibit heightened vulnerability to anti-VEGF/VEGFR cardiovascular toxicity [85], highlighting the imperative for personalized risk stratification and enhanced cardiovascular surveillance.

These paradoxical observations likely stem from the complex interplay between pathological microenvironmental variations, drug metabolism profiles, and patient genetics, highlighting the critical need for mechanistic studies.

### 3.6. Critical Analysis of Clinical Data

Current clinical studies investigating angiogenesis–AF relationships face three principal methodological limitations: first, underpowered study designs (e.g., *n* < 200), thereby limiting statistical power to detect moderate associations [77]; second, non-representative sampling through frequent exclusion of high-risk subgroups (e.g., elderly patients, those with advanced comorbidities) [87,88]; and third, single-center study designs that highten confounding risks from regional treatment protocols or genetic homogeneity [77,80].

### 3.7. Potential Causes of Contradictory Findings

The heterogeneity in findings across studies may originate from three interrelated factors: population diversity, methodological variability, and temporal dynamics. First, ethnic disparities (e.g., higher VEGF-D levels in the European population versus Asian cohorts) and variations in comorbid burden (e.g., diabetes prevalence) substantially influence angiogenesis–AF pathophysiological relationships [9,36]. Second, methodological inconsistencies—including divergent biomarker quantification approaches (e.g., ELISA versus multiplex assays) and AF detection methodologies (e.g., intermittent electrocardiogram versus continuous monitoring)—significantly contribute to outcome discrepancies [34,75]. Third, temporal fluctuations in angiogenic factor levels across AF stages (paroxysmal versus persistent) remain poorly characterized due to the predominance of cross-sectional designs, which fail to capture longitudinal dynamics [38,75].

### 3.8. Future Research Directions

To address these challenges, methodologically rigorous large-scale multicenter cohort studies are urgently needed. Standardized protocols should establish uniform definitions for angiogenesis biomarkers (e.g., VEGF isoform-specific assays) and AF endpoints (e.g., implantable loop recorder-confirmed episodes) to ensure cross-study comparability [4,75]. Prioritizing the enrollment of underrepresented populations—including non-European ethnic groups and elderly individuals (>80 years)—is essential to improve the generalizability of findings [9,28,70]. Prospective longitudinal studies incorporating serial biomarker assessments, integrated with advanced imaging modalities (e.g., cardiac MRI for fibrosis quantification), will enable precise characterization of the causal pathways and temporal relationships between angiogenesis dysregulation and AF progression.

### 3.9. Clinical Translational Potential

Emerging strategies emphasize the regulatory role of microRNAs in angiogenesis and endothelial function, identifying them as novel therapeutic targets for AF management [89]. Histone deacetylase (HDAC) inhibitors demonstrate cardioprotection by attenuating cardiac fibrosis and promoting angiogenesis, positioning them as potential AF therapeutics [90]. Additionally, therapies enhancing endothelial progenitor cell function, such as dronedarone, restore angiogenic homeostasis and may concurrently address AF and underlying endothelial dysfunction [91].

Angiogenesis biomarkers can be used in early diagnosis and risk stratification. Circulating angiogenesis-related biomarkers, such as VEGF-A, soluble VEGF receptor-2 (sVEGFR-2), and galectin-3, demonstrates clinical utility for early AF detection and risk stratification. Elevated VEGF-A levels (>450 pg/mL) in asymptomatic patients independently predict a 2.1-fold increased 5-year AF risk, irrespective of traditional risk factors [4,75]. Conversely, low sVEGFR-2 levels (<8.5 ng/mL) are prognostic for accelerated AF progression from paroxysmal to persistent forms [4]. Machine learning algorithms integrating angiogenesis markers with clinical parameters enhance the accuracy of AF prediction [88].

## 4. Therapeutic Strategies and Translational Challenges

### 4.1. Targeting Pathological Angiogenesis Inhibitor Applications

Anti-VEGF agents such as bevacizumab have demonstrated the potential to reduce atrial fibrosis by suppressing VEGF-A-driven endothelial permeability and fibroblast activation [4,84,85]. However, preclinical studies reveal that non-selective VEGF inhibition may exacerbate myocardial ischemia via impairment of compensatory angiogenesis, particularly in patients with pre-existing microvascular dysfunction [4,30]. For instance, intravitreal anti-VEGF therapies (e.g., ranibizumab) were associated with a 1.5-fold increased risk of cardiovascular events, including AF, in a pharmacoepidemiologic analysis [88].

### 4.2. Precision Interventions

Emerging strategies prioritize isoform-specific targeting to optimize the therapeutic efficacy–safety balance. Selective inhibition of VEGF-A (the pro-fibrotic isoform) while preserving VEGF-D (the protective lymphangiogenesis isoform) demonstrated attenuated atrial remodeling in rodent models [4,36]. Patient stratification using endothelial function biomarkers—such as circulating EPC levels and sVEGFR-1/sVEGFR-2 ratios—enhances treatment personalization. A recent trial identified that patients with elevated VEGF-A (>450 pg/mL) and low sVEGFR-2 (<1200 pg/mL) are optimal candidates for VEGF-A inhibitors, while those with dominant VEGF-D signaling require distinct therapeutic strategies [4,75].

### 4.3. Enhancing Compensatory Angiogenesis

Intramyocardial administration of AdVEGF-DΔNΔC, a VEGF-D isoform that promotes angiogenesis and lymphangiogenesis, has demonstrated an enhanced myocardial perfusion reserve and reduced angina symptoms in refractory angina patients [77,80]. This therapy potentiates endothelial repair mechanisms through increased nitric oxide bioavailability and oxidative stress reduction. However, its application in AF remains uncharacterized, as current trials focused on ischemic heart disease rather than arrhythmogenic substrates [77,80]. Preclinical evidence suggests VEGF-DΔNΔC may attenuate atrial fibrosis by suppressing EndMT, though its arrhythmic potential in AF requires rigorous evaluation [36,77].

MicroRNAs (e.g., miR-122) and histone deacetylase (HDAC) inhibitors represent emerging approaches to modulate angiogenic balance. Downregulation of miR-122 correlates with endothelial dysfunction in AF patients, while its exogenous overexpression restores endothelial integrity via TGF-β1 suppression [67]. HDAC inhibitors, such as valproic acid, concurrently inhibit cardiac fibrosis and stimulate compensatory angiogenesis through hypoxia-inducible factor acetylation [90].

### 4.4. Clinical Challenges in Balancing Therapeutic Effects

While VEGF inhibition reduces fibrosis, it paradoxically elevates thrombotic risk through endothelial homeostasis disruption. For instance, elevated sVEGFR-1 levels (a decoy receptor for VEGF-A) correlate with endothelial dysfunction and thromboembolic complications in AF patients, implying that excessive VEGF suppression compromises vascular integrity [4,34]. Clinical data from cancer trials underscore this therapeutic paradox: anti-VEGF agents like bevacizumab reduce fibrosis biomarkers yet double arterial thrombosis incidence versus controls [84].

## 5. Conclusions and Future Directions

Targeting angiogenesis holds therapeutic potential for AF management, potentially through modulating tissue reperfusion and local fibrosis. However, clinical evidence remains limited, with unresolved questions regarding the chronic safety and efficacy profiles of angiogenic therapies in AF. Future research should prioritize the mechanistic elucidation of angiogenesis–AF pathophysiology and conducting rigorously designed clinical trials to assess therapeutic outcomes of angiogenesis inhibitors in AF.

In conclusion, this review systematically examines the dualistic interplay between angiogenesis and atrial fibrillation, delineating its context-dependent roles in disease exacerbation and mitigation (Figure 1). Figure 1 schematizes the proposed mechanisms linking angiogenesis to AF pathogenesis, emphasizing the equilibrium between compensatory and pathological vascular remodeling. The schematic contrasts two pathways: (1) Pathological angiogenesis: Driven by dysregulated VEGF-A, FGF-2, and Ang-2 signaling, this pathway promotes atrial fibrosis, endothelial dysfunction, and electrical instability via EndMT, microvascular leakage, and inflammatory cell recruitment. (2) Compensatory angiogenesis: Mediated by Ang-1 and VEGF-D isoforms, this pathway facilitates endothelial repair, optimizes tissue perfusion, and attenuates oxidative stress.

Current evidence indicates that while excessive angiogenesis exacerbates atrial remodeling, fibrosis, and electrical dysfunction, compensatory angiogenesis counteracts ischemic and inflammatory damage through enhanced perfusion. Key angiogenic pathways—particularly the VEGF signaling—are centrally implicated in AF pathogenesis, presenting therapeutic potential. However, non-selective VEGF inhibition raises safety concerns, particularly cardiovascular toxicity risks. Future research should focus on elucidating the mechanisms of angiogenesis in AF, optimizing therapeutic strategies, and balancing pro-arrhythmic versus compensatory effects to advance personalized approaches for AF management. These efforts will advance precision medicine approaches for AF management.

## Figures and Tables

**Figure 1 biomedicines-13-01399-f001:**
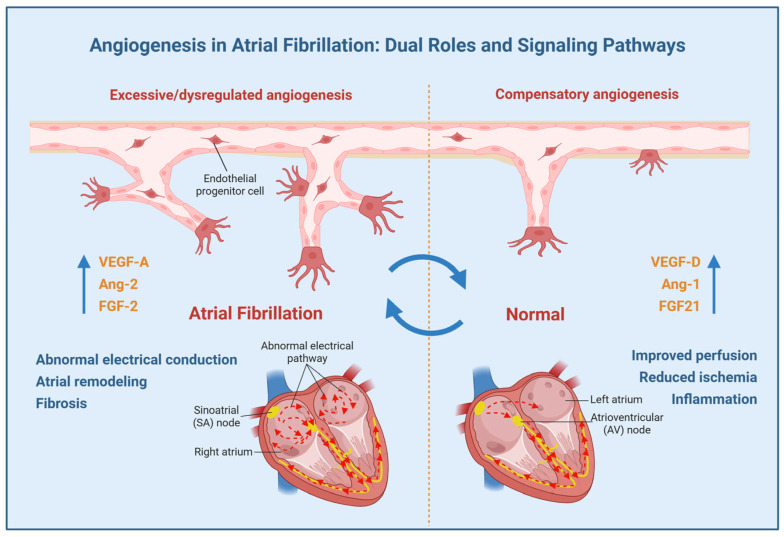
The putative role of angiogenesis in atrial fibrillation (AF). Angiogenesis in AF has a complex, context-dependent role with both pathogenic and protective effects. The net effect likely depends on the balance between pro-angiogenic and anti-angiogenic signaling pathways, modulated by the prevailing pathophysiological microenvironment.

**Table 1 biomedicines-13-01399-t001:** Key Angiogenic Signaling Pathways and Their Potential Involvement in AF Pathogenesis.

Study	Study Design	Limitations	Clinical Significance	Therapeutic Potential
Wang et al. (2019) [4]	Cross-sectional	Cross-sectional design, small sample size, exclusion of isolated AF patients without valvular heart disease	sVEGFR-2 as potential biomarker linked to endothelial dysfunction; VEGF-C unchanged, causality unresolved (cross-sectional/small sample).	Targeting VEGF/sVEGFR pathways could offer new treatment strategies for AF.
Freestone et al. (2005) [34]	Cross-sectional	Small sample size, potential confounding by comorbidities like hypertension	Elevated VEGF/Ang-2/vWF in AF patients suggest prothrombotic mechanisms; conflicting evidence (unchanged TF, Ang-1/Ang-2 imbalance), causality unresolved.	Modulating VEGF/Ang-2 pathways could potentially reduce thrombogenesis in AF, though further studies are needed.
Berntsson et al. (2019) [36]	Prospective cohort	No serial biomarker measurements, undetected AF cases	VEGF-D as potential biomarker for high-risk AF/AF-related stroke; association attenuates post NT-proBNP adjustment, causality unresolved.	Further research is needed to evaluate if targeting VEGF-D could prevent AF or stroke.
Büttner et al. (2019) [72]	Cross-sectional	Causality unclear, potential confounding by comorbidities like renal dysfunction	EGF/HB-EGF as potential AF biomarkers; causality unclear (confounders: hypertension/renal dysfunction).	Potential targets for therapeutic intervention in AF.
Sharma et al. (2025) [75]	Systematic review	Cross-sectional designs, small sample sizes, methodological heterogeneity.	VEGF-A/VEGF-D as disease activity/progression biomarkers; VEGF-C conflicts and methodological heterogeneity limit utility.	Modulating VEGF pathways could address atrial fibrosis and inflammation, offering potential therapeutic targets.
Hartikainen et al. (2017) [77]	Phase I/IIa trial	Small sample size, placebo effects from the invasive control procedure	VEGF-DΔNΔC improved perfusion reserve/angina symptoms in refractory angina; placebo similarity and small sample limit conclusions.	Offers a potential new treatment for refractory angina.
Gramley et al. (2010) [30]	Cross-sectional	No tissue analysis, inclusion of AF patients at varied stages, non-ideal controls	VEGF: up-regulation of hypoxia/fibrosis in AF pathogenesis; conflicting evidence (unchanged microvessel density, reduced KDR signaling) suggests incomplete hypoxia-driven angiogenesis.	Potential targets for therapeutic intervention in AF.
Tan et al. (2022) [70]	Meta-analysis	Small sample size, observational bias, heterogeneity across studies,	FGF-23 as potential AF risk biomarker (dose-dependent association); GDF-15 lacks significant AF link.	FGF-23 could be a potential target for AF.

Abbreviations AF: Atrial Fibrillation, VEGF: Vascular Endothelial Growth Factor, VEGF-A: Vascular Endothelial Growth Factor A, VEGF-C: Vascular Endothelial Growth Factor C, VEGF-D: Vascular Endothelial Growth Factor D, sVEGFR: soluble vascular endothelial growth factor receptor, KDR: VEGF receptor 2, Ang-2: angiopoietin-2, vWF: von Willebrand factor, HB-EGF: heparin-binding EGF-like growth factor, EGF: epidermal growth factor receptor, FGF-23: fibroblast growth factor-23, GDF-15: Growth differentiation factor-15.

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
