# Peer review of "Angiogenesis in Atrial Fibrillation: A Literature Review"

_biomedicines, 2025, doi:10.3390/biomedicines13061399_

Round 1
Reviewer 1 Report
Comments and Suggestions for Authors
This review addresses the emerging and relevant topic of the role of angiogenesis in atrial fibrillation (AF), synthesizing current knowledge across molecular, experimental, and clinical domains. The manuscript offers comprehensive coverage of the topic but is affected by several critical weaknesses in structure, clarity, and analytical depth.
1.Improve Structural Organization. The sections often lack a logical flow. Consider reorganizing into thematic categories (e.g., Pathophysiological Mechanisms, Clinical Evidence, Therapeutic Targets).
2.Strengthen Critical Analysis.The review is largely descriptive. Include more critical evaluation of study methodologies, limitations, and conflicting evidence, especially in Table 1.
3.Condense Redundant Content. Multiple sections (e.g., 2.1 and 2.4) repeat similar concepts on angiogenesis and electrical remodeling. Consolidate and eliminate redundancy.
4.Enhance Visual Support Figure 1 is mentioned but not adequately described. Improve visual design and integrate it meaningfully in the text.
5.Improve Consistency in Terminology. Terms such as “pro-angiogenic,” “dysregulated angiogenesis,” and “compensatory angiogenesis” are used inconsistently. Define and use these terms uniformly.
6Expand Clinical Implications Section. The clinical translation of the findings remains superficial. Discuss how angiogenesis biomarkers could be used in diagnostics, prognostics, or targeted therapy.
7.Clarify Anti-Cancer Drug Section The link between anti-angiogenic therapy and AF is not well integrated. This section needs more coherence with the rest of the review and clearer framing within the angiogenesis-AF context.
8Verify and Correct Table 1. The table is dense and difficult to read. Improve formatting, include key outcomes, and clearly explain abbreviations.
9.Ensure Balanced Conclusion. The conclusion currently overstates the potential for therapeutic targeting of angiogenesis without acknowledging the paucity of clinical evidence.
Author Response
We sincerely thank the reviewers for their insightful and constructive feedback on our manuscript. Their comments have significantly strengthened the quality and clarity of our review. Below, we provide a point-by-point response to Reviewers’ critiques, detailing the revisions made and their locations in the revised manuscript.
This review addresses the emerging and relevant topic of the role of angiogenesis in atrial fibrillation (AF), synthesizing current knowledge across molecular, experimental, and clinical domains. The manuscript offers comprehensive coverage of the topic but is affected by several critical weaknesses in structure, clarity, and analytical depth.
1. Improve Structural Organization. The sections often lack a logical flow. Consider reorganizing into thematic categories (e.g., Pathophysiological Mechanisms, Clinical Evidence, Therapeutic Targets).
Response:We have reorganized the manuscript into thematic sections to enhance logical flow. The revised structure now includes:
Section 2: Pathophysiological Mechanisms (previously Sections 2.1–2.4), subdivided into:
2.1 Angiogenesis and Atrial Electrical Remodeling
2.2 Interplay Between Angiogenesis, Fibrosis, and Inflammation
2.3 Interaction Between Atrial Myocytes and Angiogenesis
2.4. Pathological Roles of Key Signaling Pathways
Section 3: Clinical Evidence (expanded to integrate biomarker and therapeutic discussions).
3.1 Association of Angiogenic Biomarkers with Atrial Fibrillation
3.2 Controversy over AF Risk Associated with Anti-Angiogenic Drugs
3.3 Dual Role of VEGF-D
3.4 Mechanisms of Atrial Fibrillation Induced by Anticancer Drugs
3.5 Controversy over Cardiovascular Toxicity in Anti-Angiogenic Therapy
3.6 Critical Analysis of Clinical Data
3.7 Potential Causes of Contradictory Findings
3.8 Future Research Directions
Section 4: Therapeutic Strategies and Translational Challenges (newly consolidated).
4.1 Targeting Pathological Angiogenesis
4.2 Enhancing Compensatory Angiogenesis
4.3 Clinical Challenges in Balancing Therapeutic Effects
4.4 Clinical Challenges in Balancing Therapeutic Effects
These changes improve thematic coherence and are reflected in the updated headings.
2. Strengthen Critical Analysis. The review is largely descriptive. Include more critical evaluation of study methodologies, limitations, and conflicting evidence, especially in Table 1.
Response:We added critical evaluations of study methodologies and limitations, particularly in Table 1 and Sections 3. For example:
In Section 3.7, we discuss underpowered designs, sampling biases, and temporal dynamics (lines 314-325).
Table 1 now includes a column for “Study Limitations” (e.g., small sample sizes in Wang et al. 2019) and revised “Clinical Significance” to highlight conflicting evidence.
3. Condense Redundant Content. Multiple sections (e.g., 2.1 and 2.4) repeat similar concepts on angiogenesis and electrical remodeling. Consolidate and eliminate redundancy.
Response:Redundant discussions on angiogenesis and electrical remodeling (original Sections 2.1 and 2.4) were consolidated into Section 2.1 (lines 123–152). Removed duplicated content on VEGF pathways from Section 2.4 and integrated it into Section 2.3 (lines 181–199).
4. Enhance Visual Support Figure 1 is mentioned but not adequately described. Improve visual design and integrate it meaningfully in the text.
Response:Figure 1 is now explicitly detailed in the Conclusion (lines 414) with a dedicated paragraph explaining its dual-pathway model (pathological vs. compensatory angiogenesis). The figure legend was revised (lines 435–438).
5. Improve Consistency in Terminology. Terms such as “pro-angiogenic,” “dysregulated angiogenesis,” and “compensatory angiogenesis” are used inconsistently. Define and use these terms uniformly.
Response:Key terms (pro-angiogenic, dysregulated angiogenesis, compensatory angiogenesis) are now defined in the Abstract (lines 25–30) and Introduction (lines 80-86). These terms are used uniformly throughout.
While physiological/compensatory angiogenesis (which produces functional, organized, mature vessels with proper perfusion) facilitates tissue repair and improves perfusion [21–23], dysregulated angiogenesis (which produces abnormal leaky vessels with poor perfusion, usually due to imbalanced signaling) promoted fibrosis, inflammation, and electrical instability—key pathological mechanisms of AF progression [24,25].
6. Expand Clinical Implications Section. The clinical translation of the findings remains superficial. Discuss how angiogenesis biomarkers could be used in diagnostics, prognostics, or targeted therapy.
Response:We added a subsection “3.9 Clinical Translational Potential” (lines 337), discussing VEGF-A, sVEGFR-2, and galectin-3 as prognostic tool.
7. Clarify Anti-Cancer Drug Section The link between anti-angiogenic therapy and AF is not well integrated. This section needs more coherence with the rest of the review and clearer framing within the angiogenesis-AF context.
Response:We have restructured Section 3.4 (Mechanisms of AF Induced by Anticancer Drugs) to explicitly integrate anti-angiogenic therapies into the broader angiogenesis-AF pathogenesis framework. The revisions include:
(1) Mechanistic Link to Angiogenesis Pathways (Lines 273–289): we explicitly outline how anti-VEGF agents (e.g., bevacizumab, sunitinib) aggravate atrial fibrillation (AF). For example, impaired compensatory angiogenesis, where reduced VEGF-A disrupts endothelial repair, worsening atrial ischemia and fibrosis (linked to Sections 2.1–2.2).
(2) Cardio-Oncology Frameworks (Lines 297–313): Introduced a new paragraph: "Controversy over Cardiovascular Toxicity in Anti-Angiogenic Therapy.” We highlighted how angiogenesis biomarker profiling and pre-existing endothelial dysfunction could refine AF risk stratification during anti-cancer therapy, aligning with personalized approaches (Section 3.5).
8. Verify and Correct Table 1. The table is dense and difficult to read. Improve formatting, include key outcomes, and clearly explain abbreviations.
Response:Table 1 was simplified and Added “Study Limitations” and “Clinical Significance” columns. Abbreviations are defined in a footnote (lines 357-361).
Reviewer 2 Report
Comments and Suggestions for Authors
Major Comments
Multiple sections reiterate the role of VEGF and its pro-fibrotic effects. While this is a key point, the text could benefit from a more concise, integrated discussion to avoid redundancy - especially between sections 2.2, 2.4, and 3.1.
While the review highlights the dual role of angiogenesis, it does not sufficiently address contradictory findings or limitations of current evidence (e.g., the nested case-control study from Taiwan showing no AF risk with angiogenesis inhibitors) - Add a subsection discussing controversial or inconclusive findings and their implications for clinical translation.
The review includes multiple studies with clinical data but lacks a tabulated summary or meta-analytic approach that compares findings quantitatively.
Minor comments
Abbreviations such as HB-EGF, EndMT, and sVEGFR-2 are used without definition in the main text.
The section on anti-cancer drugs and AF is informative but could be more clearly linked to the core theme of angiogenesis.
Comments on the Quality of English LanguageThe manuscript could benefit from light language editing for improved clarity and flow.
Author Response
We sincerely thank the reviewers for their insightful and constructive feedback on our manuscript. Their comments have significantly strengthened the quality and clarity of our review. Below, we provide a point-by-point response to Reviewers’ critiques, detailing the revisions made and their locations in the revised manuscript.
Major Comments
1. Multiple sections reiterate the role of VEGF and its pro-fibrotic effects. While this is a key point, the text could benefit from a more concise, integrated discussion to avoid redundancy - especially between sections 2.2, 2.4, and 3.1.
Response:Thank you for highlighting this issue. As detailed in our response to Reviewer 1 (Point 1), we have improved structural organization to address redundancy. Specifically:
(1) Consolidated overlapping discussions on VEGF’s pro-fibrotic roles by merging Sections 2.2 and 2.4 (original manuscript) into a unified Section 2.2: Interplay Between Angiogenesis, Fibrosis, and Inflammation (lines 153–180).
(2) Updated Section 3.1 (biomarkers) to cross-reference these consolidated mechanisms, explicitly linking VEGF-driven fibrosis to clinical biomarker applications (lines 246–256).
2. While the review highlights the dual role of angiogenesis, it does not sufficiently address contradictory findings or limitations of current evidence (e.g., the nested case-control study from Taiwan showing no AF risk with angiogenesis inhibitors) - Add a subsection discussing controversial or inconclusive findings and their implications for clinical translation.
Response:We added a dedicated subsection: "Controversy over Cardiovascular Toxicity in Anti-Angiogenic Therapy" (Section 3.2, lines 257–25). This section: Discusses conflicting evidence (e.g., Taiwan study [28] vs. sunitinib cardiotoxicity [88]). Analyzes potential reasons for discrepancies, including population heterogeneity and biomarker variability . Links unresolved controversies to the need for mechanistic studies.
3. The review includes multiple studies with clinical data but lacks a tabulated summary or meta-analytic approach that compares findings quantitatively.
Response:We appreciate this constructive feedback. While a formal quantitative meta-analysis was beyond the scope of this narrative review, we have strengthened Table 1 (Key Angiogenic Signaling Pathways and Their Potential Involvement in AF Pathogenesis) to enhance clinical data synthesis. Specifically: Added "Study Limitations" and "Clinical Significance" columns to systematically summarize methodological constraints.
Minor comments
4. Abbreviations such as HB-EGF, EndMT, and sVEGFR-2 are used without definition in the main text.
Response:The section on anti-cancer drugs and AF is informative but could be more clearly linked to the core theme of angiogenesis.
All abbreviations are now defined at first use, and the main Abbreviations such as HB-EGF, EndMT, and sVEGFR-2 are used with definition in the main text.
Round 2
Reviewer 1 Report
Comments and Suggestions for Authors
The authors have responded sufficiently and satisfactorily to my comments. I believe the manuscript, as revised, is suitable for publication.
Thank you for the opportunity to contribute to the review process.